# Rapid Biodistribution of Fluorescent Outer-Membrane Vesicles from the Intestine to Distant Organs via the Blood in Mice

**DOI:** 10.3390/ijms25031821

**Published:** 2024-02-02

**Authors:** Béatrice Schaack, Corinne Mercier, Maya Katby, Dalil Hannani, Julien Vollaire, Julie Suzanne Robert, Clément Caffaratti, Françoise Blanquet, Olivier Nicoud, Véronique Josserand, David Laurin

**Affiliations:** 1CNRS, UMR 5525, VetAgro Sup, Grenoble INP, TIMC, University Grenoble Alpes, F-38000 Grenoble, France; beatrice.schaack@ibs.fr (B.S.); corinne.mercier@univ-grenoble-alpes.fr (C.M.); katbymaya@gmail.com (M.K.); dalil.hannani@univ-grenoble-alpes.fr (D.H.); clement.caffaratti@gmail.com (C.C.); francoise.blanquet@univ-grenoble-alpes.fr (F.B.); 2CEA, CNRS, IBS, University Grenoble Alpes, F-38000 Grenoble, France; 3INSERM U1209, CNRS UMR5309, Institute for Advanced Biosciences, University Grenoble Alpes, F-38000 Grenoble, France; julien.vollaire@univ-grenoble-alpes.fr (J.V.); olivier.nicoud@univ-grenoble-alpes.fr (O.N.); veronique.josserand@univ-grenoble-alpes.fr (V.J.); 4Etablissement Français du Sang, Département Scientifique Auvergne Rhône-Alpes, F-38000 Grenoble, France; julie04robert@gmail.com

**Keywords:** biodistribution, extracellular vesicles, miRFP713, outer membrane vesicles (OMVs), live imaging, microbiota

## Abstract

A cell’s ability to secrete extracellular vesicles (EVs) for communication is present in all three domains of life. Notably, Gram-negative bacteria produce a specific type of EVs called outer membrane vesicles (OMVs). We previously observed the presence of OMVs in human blood, which could represent a means of communication from the microbiota to the host. Here, in order to investigate the possible translocation of OMVs from the intestine to other organs, the mouse was used as an animal model after OMVs administration. To achieve this, we first optimized the signal of OMVs containing the fluorescent protein miRFP713 associated with the outer membrane anchoring peptide OmpA by adding biliverdin, a fluorescence cofactor, to the cultures. The miRFP713-expressing OMVs produced in *E. coli* REL606 strain were then characterized according to their diameter and protein composition. Native- and miRFP713-expressing OMVs were found to produce homogenous populations of vesicles. Finally, in vivo and ex vivo fluorescence imaging was used to monitor the distribution of miRFP713-OMVs in mice in various organs whether by intravenous injection or oral gavage. The relative stability of the fluorescence signals up to 3 days post-injection/gavage paves the way to future studies investigating the OMV-based communication established between the different microbiotas and their host.

## 1. Introduction

An increasing number of diseases have been associated with changes in the gut microbiota composition. These changes concern a wide range of pathologies, from diseases such as inflammatory bowel disease or the metabolic syndrome to cancer or pathological processes observed in the brain [1,2]. Data have suggested an alteration of the gut microbiota composition as a part of pathogenesis. For example, bacteria are increasingly recognized as key players in the tumorigenesis of certain cancers, while numerous studies also confirm the role of the intestinal microbiota in modulating therapeutic responses, particularly to cancer immunotherapies [3,4]. However, the relationship between changes in the intestinal microbiota and the observed pathological processes remains complicated to understand and analyze. It is particularly important to identify and evaluate the mediators, which are involved. A plausible key mechanism for communication between organs altered by cancer, the brain or responses to therapeutic treatments, and the bacteria of the gut microbiota is communication via the secretion of bacterial extracellular vesicles (BEVs).

The release of extracellular vesicles (EVs) is a mechanism that has been preserved in both multi- and unicellular organisms (Gram-negative and Gram-positive bacteria, Archaea, Yeast and Protozoa) during evolution [5,6]. Whatever the type of cell they originate from, EVs constitute universal tools for inter- and intra-kingdom communication [5]. These communication enables complex interactions within the microbiota but also with the host. The internal cargo of BEVs is protected from degradation in the extracellular environment by enzymes. Moreover, BEVs contain pathogen-associated molecular patterns and can initiate inflammatory signaling pathways in host cells [7]. They may therefore play an essential role in the pathogenesis of various diseases. In addition to their role in pathogenesis, BEVs could also be of interest for diagnostic or therapeutic interventions [8].

Gram-negative bacteria produce a sub-type of EVs dubbed outer membrane vesicles (OMVs), with diameters ranging from 20 to 400 nm. Like any cell, Gram-negative bacteria are delineated by a plasma membrane known as their inner membrane. In addition to this membrane, Gram-negative bacteria also exhibit an outer membrane (OM) that consists of an inner leaflet of phospholipids and an outer leaflet mainly composed of lipopolysaccharides (LPS). This OM is also made of transmembrane proteins including porins such as the outer membrane porin A (OmpA), which is one of the main proteins of the *Escherichia coli* OM. These porins play multiple roles, including shielding the bacterium or facilitating its communication. Interestingly, OmpA, which has been shown to be integrated within the OMV membrane, has also been characterized by its high level of immunogenicity and its usefulness for bioengineering OMV content [9]. The inner content of OMVs has been shown to contain numerous bioactive molecules such as enzymes, toxins, antigenic determinants, nucleic acids, and metabolites [10].

In vivo, OMVs produced by Gram-negative bacterial species of the intestinal flora are released into the intestinal lumen and affect their environment well beyond their parental cells [11,12]. Intestinal microbiota-derived OMVs can be phagocytosed by immune cells within the lamina propria [13]. The vast majority of studies on gut microbiota BEVs have focused on pathogenic bacteria and/or pathological contexts associated with permeability or fragility induced by inflammatory situations of the intestinal barrier. However, mounting evidence indicates that intestinal BEVs pass through mucosal barriers under healthy conditions. Indeed, we previously reported the presence of Enterobacteriaceae-derived Gram-negative BEVs (OMVs) in transfusion blood products prepared from qualified healthy donors [14]. A recent study focusing on BEVs in circulating blood confirmed our observations, showing that the main origin of these vesicles is the gut microbiota [15]. This study also showed that the intestinal barrier becomes more permeable with age. These observations suggest that in the absence of dysbiosis, OMVs 1/can cross the intestinal epithelium and the vascular endothelium to reach sites beyond the gastro-intestinal tract and 2/can travel to distant organs via blood circulation. It has been proposed that in order to cross the intestinal epithelium or the vascular endothelium, OMVs use two distinct pathways, either the paracellular (between cells) or the transcellular (through cells) pathways [11]. Lee and collaborators [16] used *Paenalcaligenes hominis* extracellular vesicles to follow from the gut to the brain the distribution of OMV traces as LPS molecules. However, so far, in healthy conditions, the translocation of OMVs from the gut to the periphery has never been directly observed nor analyzed.

Thus, in this study, we used healthy mice to track the journey of bacterial OMVs, in particular their translocation, and identify the organs where these OMVs have spread after oral administration. To this aim, we engineered a non-pathogenic *Escherichia coli* strain capable of producing OMVs containing an easily detectable and quantifiable fluorescent protein, namely miRFP713 emitting in the near infrared, which is a modified version of the bacteriophytochrome from *Rhodopseudomonas palustris* used for non-invasive and deep tissue in vivo imaging [17]. The selected *E. coli* model is REL606, which is the B ecotype strain chosen by R.E Lenski as the ancestor of the long-term evolution experiment (LTEE: more than 30 years of bacterial evolution, https://the-ltee.org/). It is a commensal bacterial strain isolated from human microbiota that does not contain any phage or plasmid and its genome has been sequenced and assembled [18]. OmpA is the major protein described in OMVs. Its signal peptide (OmpA SP) has been successfully used to address recombinant proteins to the OMVs of *Bacteroides* [19]. For this study, REL606 *E. coli* were transformed with an expression plasmid carrying the open-reading-frame of miRFP713 placed downstream of the *lac* promoter and in frame with the OmpA signal peptide. This construct ensured the export of the recombinant protein into the OMVs secreted by the transformed bacteria [20].

To prove that OMVs can cross cell and tissue barriers, we purified and characterized OMVs secreted from miRFP713-expressing-REL606, referred to as miRFP713-OMVs, and compared them with wild-type OMVs secreted by the REL606 strain. MiRFP713-OMVs were administrated to mice using intravenous injection or oral gavage, since it had previously been reported that OMVs reach the gut after oral gavage [21]. Intravenous injection of OMVs was performed as a back-up experiment should the fluorescence signal following gavage be too low.

We finally report on the quantified levels of in vivo and ex vivo fluorescence while monitoring the gradual diffusion of miRFP713-OMVs from the gastro-intestinal track to the entire mouse.

## 2. Results

### 2.1. Biliverdin Enhances the Fluorescence of miRFP713 Expressed in REL606

Used at the final concentration of 25 µM in bacterial cultures, biliverdin (BV) had been described as a miRFP713 cofactor that greatly enhances the fluorescence intensity [22]. To optimize the fluorescence emitted by miRFP713 expressed in REL606 (Figure 1A–C), we tested the effect of the addition of BV during the growth, over 20 h, of the three bacterial clones selected for their high intensity of fluorescence (Figure 1D). First, 2 mL of LB broth supplemented with 100 µg/mL Amp was inoculated with each of these three identified clones. The 2 mL cultures were placed at 37 °C, under 180 rpm, and the emitted fluorescence of the multiplying bacterial cells was measured at an emission wavelength of 680 nm during 100 ms, respectively, at 0, 0.5, 1, 2, 4, and 16 h after the addition of BV. The emitted fluorescence of the three cultures increased with exposure to BV and fluorescence intensity; i.e., 100 units were detected when cells had been exposed to BV for 20 h. Importantly, the fluorescence intensities of the three tested clones were superimposable (Figure 1E). Clone A (Figure 1D,E) was thus arbitrarily chosen for the experiments reported thereafter.

Given that the production of OMVs (Figure 2A) had been optimized in the laboratory for 250 mL bacterial cultures [14,23], clone A was identified in the 96-well plate (Figure 1D) as providing the highest level of fluorescence; thus, it was amplified stepwise, and its fluorescence spectra was recorded. The excitation spectrum of miRFP713 ranged from 620 to 660 nm with a peak at 640 nm, while the emission spectrum ranged from 640 to 700 nm with a peak at 660 nm, which corresponded to 100 units of fluorescence (Appendix A), thus confirming that the level of fluorescence obtained from a 250 mL bacterial culture in the presence of 25 µM BV was almost equivalent to that obtained from a 2 mL culture volume with 25 µM BV. Finally, the linearity of miRFP713 fluorescence emitted by the selected clone A was verified by stepwise dilutions in PBS of the 250 mL overnight culture in the presence of 25 µM BV and further measurement of the fluorescence emitted by the diluted bacterial cells (Appendix A). OMVs would therefore need to be prepared using overnight cultures (minimum of 16 h of culture) in the presence of 25 µM BV since BV increased the fluorescence emitted by the bacterial OMVs more than eight-fold.

### 2.2. Both the REL606- and the miRFP713-Expressing REL606 E. coli Strains Secrete Homogenous Populations of OMVs with Close Radii

To further characterize the OMVs secreted by the REL606 wild-type strain versus the miRFP713-expressing REL606 strain, we first verified that the detected fluorescence came from OMVs and not from aggregated proteins released from bacterial cells nor diffused from OMVs. OMVs prepared from the miRFP713-expressing REL606 strain were thus loaded at the top of a sucrose density gradient (from 0 to 60%) and then centrifuged for 3 h at 150,000× *g* with the collected fractions analyzed by fluorescence. Positive signals were detected only in the lightest fractions of the gradient (both the 20% and the 10% sucrose fractions), thus confirming that miRFP713 was associated with the lipid vesicles known to float to the lightest fractions of the gradient. OMVs were produced from 250 mL of each bacterial culture, purified according to the protocol previously reported [23] and schematized on Figure 2A; their radius was measured by dynamic light scattering (DLS). The two bacterial stains produced homogenous populations of OMVs (one single peak per strain). The mean radius of the REL606 OMVs was ~56 ± 3 nm, while that of the miRFP713-expressing REL606 OMVs was ~41 ± 3 nm (Figure 2B). Observation of REL606 OMVs by transmission electron microscopy revealed a regular globular shape for the OMVs, the three examples depicted in Figure 2C exhibited diameters of 67 nm (left panel), 57 nm (central panel) and 105 nm (right panel).

### 2.3. OMVs from REL606 and miRFP713-Expressing REL606 Contain Variable Amounts of OmpA

To investigate the proteins contained in the OMVs from the two REL606 strains, their SDS-PAGE pattern was first observed after Coomassie blue staining. Various bands with variable relative intensities were observed on the two types of OMV protein patterns. However, the major band of the two types of OMVs was at ~30 kDa (Figure 2D, black arrow head). This apparent molecular weight being suggestive of OmpA, its presence was confirmed in the two types of OMVs by immunoblot: two bands of, respectively; ~35 kDa and ~27 kDa were detected in the two types of OMVs (Figure 2E). However, their respective intensities were inverted: the strongest signal was at ~35 kDa (pre-OmpA) in REL606 OMVs and at ~27 kDa (OmpA) in miRFP713-OMVs.

Analyses of size, morphology and protein content confirmed that we have succeeded in producing OMVs similar in structure to native OMVs when transforming the bacterial cells with the plasmid encoding the recombinant miRFP713 protein.

### 2.4. OMVs Secreted by miRFP713-Expressing REL606 Interact with Human Blood Monocytes

Given that previous work in the laboratory demonstrated that REL606 OMVs interact primarily with human blood monocytes rather than any other blood cell type [14], we investigated if OMVs that had incorporated the miRFP713 were altered for this property. Thus, miRFP713-OMVs were incubated for 24 h with human Peripheral Blood Mononuclear Cells at a 10,000 OMV-miRFP713/cell ratio. Cells cultured for the same time in medium were used as control. After labeling with fluorescent anti-CD14 antibodies, the signal was gated on the CD14^+^ monocytes, revealing that 76.7% of the monocytes had acquired an intense miRFP713 fluorescence signal corresponding to 960 units as compared with the 173 units detected for the control monocytes (Figure 3). These results confirmed that the presence of miRFP713 in the OMVs secreted by the engineered REL606 *E. coli* strain could still be captured by human monocytes.

### 2.5. Following Injection into the Tail Vein, miRFP713-OMVs Distribute Broadly into the Whole Mouse Body Where They Can Be Detected in Various Organs up to 24 H Post-Injection

To investigate the feasibility of tracking the biodistribution of miRFP713-OMVs in mice by in vivo animal imaging, the linearity of the fluorescence detection and the detection limit were evaluated using 10 µL samples: pure, 1/2th, 1/4th, 1/8th, 1/16th, 1/32th, 1/64th, 1/128th, and 1/256th dilutions of miRFP713-OMVs (in murine plasma) were first loaded onto the LUMINA III imaging device (PerkinElmer, Waltham, MA, USA). As expected, the measured fluorescent signal increased in a proportional way accordingly to the concentration of fluorescent OMVs (Appendix A), but the intensity was limited to ensure observation in live animals. Fortunately, this signal was sufficient for the observation of fluorescence in isolated organs. Thus, two hundred microliters of miRFP713-OMVs (quantity adjusted to 100 μg of proteins) was injected intravenously into the tail of three mice. Blood samples drawn from tip tail scarification, immediately after OMV injection then at 1, 2.5, 5, and 24 h post-injection (p.i.) showed a peak of fluorescence in the plasma 1 h p.i., which was followed by a decrease in the fluorescence signal until 5 h p.i., when the signal returned to background level (Appendix A). Blood signal decreased following two phases (a rapid and a slow), and from 1 h after injection, no more significant signal was detected in blood. Blood fluorescence signals were set using a two-phase decay model (using Graphpad Prism 9.3). The corresponding half-lives were 2.1 min for the diffusion + elimination first phase and 17.8 min for the slow elimination phase.

Despite a strong auto-fluorescent signal from the fur, the brain and the stomach of the injected mice (Figure 4A, T = 0), in vivo detection of the fluorescence signal in the three injected mice revealed a strong positive signal from the bladder and the lower urinary tract 1 h p.i. (Figure 4A, mouse “supine” position). Together, these results suggested (1) a rapid distribution of miRFP713-OMVs over the whole mouse organism through the blood circulation following their injection into the tail vein, and (2) that complete elimination of miRFP713-OMVs from the organism would take at least 5 h.

To refine this distribution, several organs (the heart, brain, lungs, gut, uterus with its ovaries, the liver, pancreas, adrenal glands, some peritoneal fat, the spleen, kidneys, quadriceps muscles, and a piece of skin from the back) were harvested from three of the injected mice at 5 h and 24 h p.i. compared with the same organs from a non-injected mouse. Despite the fact that a strong signal of auto-fluorescence was detected mainly in the lungs, brain, fat, uterus/ovaries, the skin, liver, and pancreas of the non-injected mouse (Figure 4C “non-injected”), significant positive signals were detected ex vivo in the skin, gut, liver, and the fat 5 h p.i. These fluorescence signals were no longer detectable 24 h post-injection.

The results of this experiment thus (1) confirmed that miRFP713-OMVs can diffuse through the circulating blood to distant organs; (2) demonstrated that these fluorescent OMVs gather in various organs including those that exhibit their own microbiota flora (the gut and the skin) 5 h p.i., where they might deliver information to the local flora; and (3) suggested that OMVs may have potential effects on various host physiological pathways since the fluorescence of miRFP713-OMVs was detected broadly in various organs—not only in the organs involved in the elimination of wastes (the bladder, the intestine, or the liver).

To complete these observations, we also individualized cells from the harvested spleens and analyzed the presence of their miRFP713 signal (Figure 4E). We focused our analyses on the monocyte subsets M1 and M2 as well as conventional myeloid dendritic cells (cDCs) types 1 and 2 (cDC1 and cDC2). Fluorescent signals were detected from both types of monocytes 5 h p.i., and even more intense signals were detected 24 h p.i. MiRFP713 intense fluorescence signals were detected from both cDC1 and cDC2 cells5 h p.i. with a decrease in the fluorescence signals 24 h p.i. As DCs are sensors of danger signals and constitute the upstream cells that control adaptive immune responses, expression of their activation marker CD80 as well as their antigen-presenting MHC class II complex was measured (Figure 4F,G). In both types of DCs, CD80 and MHC class II expression followed the same trend with an increased expression of CD80 5 h p.i. followed by lower fluorescence signals 24 h p.i. Although we cannot exclude the possibility that the DCs, following their activation, had been mobilized to other organs such as lymph nodes, or that the miRFP713 molecules captured by the cells from OMVs have been degraded to process antigens, these results clearly demonstrated that OMVs released from the gut microbiota can interact with major actors of the immune system as soon as reaching the circulating blood, thus potentially modifying the quality and the intensity of immune responses.

### 2.6. Oral Gavage of miRFP713-OMVs Confirmed Their Possible Bio-Diffusion to Distant Organs and Their Persistence for Several Days in Mice, Thus Revealing Their Resistance to the Gastric Environment

To investigate the feasibility of miRFP713-OMVs fluorescence detection, the linearity of the detected intensity of fluorescence and the detection limit were first evaluated using 10 µL samples with various dilutions (Pure, 1/2th, 1/4th, 1/8th, 1/16th, 1/32th, 1/64th, 1/128th, 1/256th in murine plasma) loaded onto the LUMINA III imaging device (PerkinElmer). Despite low signals, the measured fluorescent signal increased proportionally to the concentration of fluorescent OMVs (Appendix A). Low in vitro detectability meant that poor fluorescence detection sensitivity was expected in vivo in mice but sensitivity would be sufficient for ex vivo imaging.

To track the biodistribution of OMVs administered per os in mice, 200 µL of miRFP713-OMVs (quantity adjusted to 100 μg of proteins) was gavaged into each of six female BalB/C mice at Day 0, Day 3 and Day 5. When measured using the LUMINA III imaging device, quantification of the emitted fluorescence of the plasma samples prepared from blood samples drawn from tip tail scarification of these mice before and at regular times following gavage and up to 24 h showed that (1) the highest level of fluorescence was detected as soon as 1 h post-gavage (p.g.), demonstrating an almost instantaneous diffusion of the fluorescent vesicles into the blood circulation post-gavage (p.g.) (Figure 5B); (2) the detected fluorescence remained moderate (1.5 unit at the most compared with a threshold value set up at 0.5 unit, Figure 5B,C); (3) soon after gavage, the fluorescence detected in circulating blood declined during the first 5 h p.g. and stabilized at 1 unit until 24 h p.g. (Figure 5C). Importantly, comparison of the ratio of fluorescence detected in the blood of one mouse to that of the OMVs originally gavaged revealed that more than 83% of the gavaged miRFP713-OMVs traveled from the intestine to the blood (gavaged miRFP713-OMVs: 45 U × 0.2 mL = 9 U, as compared with fluorescence detected in the blood: 1 mL of plasma × 7.5 U = 7.5 U). Quantification of the fluorescence detected in blood samples 5 h p.g. using a fluorimeter confirmed that the three gavaged mice exhibited higher levels of fluorescence than the three mock-gavaged mice (Figure 5D). However, this measurement revealed a variation in the level of fluorescence between the three gavaged mice with the mean level of fluorescence being 7 units for two of the three gavaged mice and only 3.5 units for the third mouse as compared with levels of fluorescence that varied between 0.5, 1, and 2 units for the three control mice (Figure 5D). These results indirectly demonstrated that miRFP713-OMVs can diffuse efficiently from the gastro-intestinal tract into the blood system in less than one hour and confirmed that the miRFP713-OMVs’ stability in the circulatory system can last 24 h.

Measurement of in vivo fluorescence emitted by the mock-gavaged mice confirmed a high level of auto-fluorescence in the animals, in particular in their fur (Figure 5A, Day 0, T = 0). Nevertheless, following the primary gavage, a positive fluorescent signal was detected in the stomach of the animal 1 h p.g. (Figure 5A, D0-1 h) and in the bladder at 2.5 h p.g. (Figure 5A, D0-2.5 h). Intriguingly, the fluorescent signal was not strong enough to be detected 5 h p.g. (Figure 5A, D0-5 h). Consecutively to the second gavage, at Day 3, the fluorescent signal was first detected in the stomach with a faint signal in the upper gut (Figure 5A, D3). A fluorescence signal was detected in the stomach and the upper part of the intestine 1 h p.g. (Figure 5A, D3-1 h), which was followed by a continuous signal in the stomach 2.5 h and 5 h p.g. (Figure 5A, D3-2.5 h and D3-5 h). At Day 5 after the original gavage, a fluorescent signal was still detected in the stomach (Figure 5A,E). The fluorescence signal detected following the third gavage at Day 5 remained in the stomach 1 h p.g. and was no longer detected (Figure 5A, D5, 24 h).

To refine this analysis, the kinetics of fluorescence was monitored in vivo at various time points in the stomach, the intestine, the liver and the skin from the 1st gavage and up to 6 days following the 1st gavage (which happened 24 h after the 3rd gavage) (Figure 5F–I). In the stomach, the fluorescence ratio reached 2 units immediately after the 1st gavage and declined steadily over the 3 days following the gavage; it increased again up to 2.5 units after the 2nd gavage, remained there until 5 days from the 1st gavage and declined only after the 3rd gavage to reach a level close to the threshold 6 days after the 1st gavage (Figure 5F). In contrast to this steady decrease in fluorescence in the stomach following gavages, the ratios of fluorescence detected in the intestine (Figure 5G), the liver (Figure 5H) and the skin (Figure 5I) revealed a common pattern: though remaining globally low, they increased progressively after the 1st gavage, decreased after the 2nd gavage and remained globally stable at ~1.5 unit after the 3rd gavage.

To refine this analysis, the fluorescence levels were monitored in various organs (the heart, lungs, brain, uterus + ovaries, the adrenal glands, the peritoneal fat, the spleen, kidneys, skin from the back, the quadriceps muscles, the liver, pancreas, stomach, intestine, caecum, and colon) dissected either 5 h after the 1st gavage or 24 h following the 3rd gavage. The results showed (1) significant signals in the skin, the stomach and the digestive system 5 h after the first gavage (2) and a significant signal only in the skin 24 h after the 3rd gavage (Figure 5E). These results suggested a gradual diffusion of the fluorescent OMVs from the gastro-intestinal track (through which most of the OMVs would be nevertheless eliminated) to the most distant organs such as the skin within a 5-day time period.

Together, these results revealed that the fluorescence of miRFP713-OMVs administered per os to mice (1) diffuses gradually from the stomach to the rest of the gastro-intestinal track where it can be positively detected up to 6 days after the 1st gavage in the stomach, the intestine, the caecum, the colon as well as the glands associated with the digestive system (the liver and the pancreas) and (2) concentrates in the skin 6 days after the 1st gavage.

## 3. Discussion

Our recent results have demonstrated the presence of bacterial vesicles in the circulating blood of healthy human blood donors [14]. The ultimate goal of this study was to track the transport of OMVs from the intestine to the other organs of the host in a mouse model. To do so, we used deep organ imaging to follow the distribution of OMVs secreted from engineered *E. coli* to express a plasmid that encodes a near-infrared fluorescent protein, miRFP713. To this aim, we transformed REL606 *E. coli* with a plasmid designed to allow the expression of miRFP713, the export of the fusion protein to the bacterial periplasm being ensured by the OmpA signal peptide (Figure 1A). Although a satisfying efficacy of transformation was achieved (25.10^3^ bacteria/µg of DNA), the fluorescence of the transformed colonies, as tested in a 96-well plate after one night of culture (time necessary to allow the expression of miRFP713) was very low, and only 3 clones out of the 96 tested provided a fluorescence of 12 units (Figure 1B). To compensate this low level of fluorescence, an addition of 25 µM BV to the bacterial cultures was tested: a 16 h incubation time period proved to be the most efficient for boosting the fluorescence emission of miRFP713 (Figure 1E). BV is a small (582.6 g/mol) tetrapyrrolic product, which thus should pass through membranes easily. Although its mechanism of action is not completely understood, it has been proposed that (1) the excited state energy of BV would be transferred to miRFP713 by resonance energy transfer, thus boosting the protein’s fluorescence emission and (2) BV would serve as a cofactor helping miRFP713 achieve its proper folding and maturation, thus increasing its fluorescence intensity [19,24]. To produce a sufficient quantity of fluorescent OMVs from transformed bacteria, BV was routinely added during the entire length of the 16 h bacterial culture. As previously reported, recombinant proteins are sent to the bacterial periplasm after stationary growth [24]. The results reported herein now show that the recombinant miRFP713 proteins are able to reach the OMVs when expressed in fusion with the OmpA SP.

The biochemical and biophysical characterization of the OMVs produced by miRFP713-expressing REL606, as compared to those produced by REL606, revealed several interesting features. Both the excitation and the emission spectra of miRFP713, when expressed by the 250 mL bacterial cultures (Appendix A), were unexpected. The excitation spectrum indeed peaked at 640 nm (Appendix A) when it was expected at 690 nm [25], and the emission spectrum peaked at 660 nm (Appendix A) when it was expected at 713 nm [25]. The theoretical setting for miRFP713 detection should thus be 690/713 nm for excitation and emission, respectively. Experimental testing showed no detectable signal at these wavelengths, while the 620/670 nm setting provided a significant signal. This could be interpreted as a modification of the protein’s optical properties related to its local environment in OMVs. DLS measurements showed that each of the REL606- and miRFP713-expressing-REL606 OMV populations constituted single populations (unimodal distribution) with peak radii of ~56 nm and ~41 nm, respectively (Figure 2B). To increase the number of produced OMVs, both types of OMVs were produced during the stationary phase of the respective bacterial cultures. These radii values were double when compared to that of REL606 OMVs produced during the exponential phase (~21 nm) [23]. Therefore, the OMVs from exponential and stationary phases differed. It has indeed been well characterized that growth conditions induce OMVs heterogeneity on multiple levels [26].

After the addition of BV to the cultures, the fluorescence of miRFP713-OMVs was sufficient to inject or administer them per os to mice. First, the caudal intravenous injection of these fluorescent miRFP713-OMVs was followed by an intense signal in the blood (Figure 4A) and the bladder 1 h p.i. (Figure 4B). These results suggest that a substantial fraction of the OMVs was rapidly directed to the kidneys and then the bladder for elimination with urine upon injection into the caudal vein. However, the fluorescence levels detected ex vivo in spleen cells 5 h p.i. (Figure 4E) in the three injected mice (Figure 4C,D) and the fluorescent signal detected in human monocytes incubated in vitro with miRFP713-expressing-REL606 OMVs (Figure 3) suggested that OMVs diffuse very rapidly into the blood circulation following injection into the tail vein. Once in the blood, they interact primarily with blood monocytes in a similar way as REL606 OMVs were shown to interact with human blood monocytes (our recent publication [14]. In addition, miRFP713-OMVs are immunogenic due to the presence of LPS in their membrane [23]. The uptake of intravenously administered OMVs by monocytes and other myeloid cells such as conventional DCs type 1 or 2 (cDC1 and cDC2) was confirmed by miRFP713 fluorescence analyses on these spleen cells (Figure 4). On the one hand, the miRFP713 signal was maintained for 24 h in monocytes. In DCs, on the other hand, the signal was stronger at 5 h than 24 h post-intravenous injection, suggesting that the cells that had taken up the OMVs had either degraded the miRFP713 protein or had migrated to another site.

Up to 83% of the fluorescence from the gavaged miRFP713OMVs were detected in the circulating blood 5 h p.g. (Figure 5D), revealing that miRFP713-OMVs efficiently and rapidly translocate from the intestinal tract to the circulating blood. Furthermore, fluorescence was detected for up to 6 days after the original gavage in the periphery of the mouse organism, in particular the skin (Figure 5B), which suggests that at least a fraction of these fluorescent OMVs exhibit a long half-life and would be relatively stable. This is in agreement with our results that revealed a single homogenous peak when miRFP713-OMVs were analyzed by DLS two weeks after preparation (storage at 4 °C between the moment of their preparation and their analysis by DLS).

## 4. Materials and Methods

### 4.1. Bacterial Strains Used for the Preparation of OMVs

The *E. coli* REL606 already used in Laurin et al. [23] (https://the-ltee.org/), was kept on the long term at −80 °C in 50% glycerol and used for the preparation of competent cells stored in aliquots of 100 µL at −80 °C in 10% glycerol. The 63 bp OmpA-SP (encoding MKKTAIAIAVALAGFATVAQA) followed by the miRFP713 948 bp coding region was introduced between the EcoRI and HindIII cloning sites of the pRSET vector. For each production of OMVs, an aliquot of REL606 competent cells was thawed on ice and gently mixed with 100 ng of the 4.4 Kb miRFP713-pRSET (Figure 1A), the construction of which was sub-contracted and the sequence verified by GeneArt (Thermofisher). The mixture was incubated for 30 min on ice before being heat shocked for 40 sec at 42 °C. Then, 500 µL of Super Optimal Broth (SOC: 2% tryptone, 0.5% yeast extract, 10 mM NaCl, 2.5 mM KCl, 10 mM MgCl_2_, 10 mM MgSO_4_, and 20 mM glucose) was added, and the bacterial cells were incubated for 30 min at 37 °C before being plated on Luria Bertani (LB) agar plates containing 100 µg/mL ampicillin (Amp) and further incubated overnight at 37 °C. Previous studies having shown that the RFP protein requires 12 h to be expressed in the presence of Amp [13]; all the liquid cultures reported in this article were incubated in LB broth for 16 h at 37 °C, 180 rotations per minute (rpm), leading to absorbances (A) at 600 nm greater than 1.5. The colonies recovered from the transformation experiment were tested after growth in a 96-well black plate (flat bottom, Thermofisher) for their emission of fluorescence. Each colony was inoculated using a tip into a well that contained 200 µL of LB supplemented with 100 µg/mL Amp and 25 µM BV. The plate was incubated for 16 h at 37 °C, 180 rpm, and the fluorescence of each well was measured using a Varioskan plate reader (excitation wave length: 620 to 660 nm, emission wave length: 680 nm; Thermofisher). The bacterial cells of the 3 wells providing the highest fluorescence were selected for further scaling up of their culture.

### 4.2. Scaling up the Cultures of miRFP713-Expressing-REL606 Bacteria

Previous experiments performed in the laboratory [23] demonstrated that a minimum volume of 250 mL of REL606 culture is required to recover a sufficient amount of OMVs for further experiments. Each clone identified in 96-well plates as emitting the highest levels of fluorescence was thus inoculated into 8 mL of LB supplemented with 100 µg/mL Amp in a 50 mL flask and incubated for 16 h at 37 °C, 180 rpm. Each 8 mL culture was subsequently used to inoculate 250 mL of LB supplemented with 100 µg/mL Amp (in a 1.5 L Erlenmeyer flask). The fluorescence spectra of miRFP713 expressed by the bacteria were recorded using a Varioskan fluorimeter (Thermofisher) after the addition of 25 µM BV, varying the excitation wave length from 620 to 660 nm while the emission wavelength was set at 680 nm for 100 ms.

### 4.3. Production of OMVs from REL 606 versus miRFP713-Expressing-REL606

The REL 606 strain (stored at −80 °C) and the miRFP713-expressing-REL606 fluorescent clone (kept at 4 °C) were revived overnight at 37 °C, 180 rpm, in 8 mL cultures of LB broth supplemented with 100 µg/mL Amp as a pre-culture. The 8 mL pre-cultures were further introduced into 1.5 L Erlenmeyer flasks containing 250 mL of LB supplemented with 100 µg/mL Amp and 25 µM BV. The cultures were incubated for 16 h at 37 °C, 180 rpm. The final A_600_ was measured, and the bacterial cells were pelleted by a 45 min centrifugation at 3500 rpm (2058× *g*), 10 °C. The supernatant was filtered through a 0.2 μm filter using a vacuum pump and further ultracentrifuged at 41,000 rpm (100,000× *g*) for 1 h, at 4 °C, in 25 mL ultracentrifugation tubes (Beckman, Brea, CA, USA). After ultracentrifugation, a small volume of supernatant was left at the top of the pellet to avoid the disruption of this fragile and invisible OMV pellet. The final volume left in the tubes after the last ultracentrifugation was used to disperse the OMV pellets, and the final OMV solutions were kept at 4 °C pending further experiments. Before analysis, the samples were filtered again (0.2 μm) in order to remove any potential OMV aggregate.

### 4.4. Quantification of the OMV Contents in Lipids by 8-Anilinonaphthalene-1-Sulfonic Acid (ANS) Assay

ANS (Sigma, St. Louis, MO, USA) is a lipophilic reagent that incorporates naturally into membranes and fluoresces at 512 nm when excited at 350 nm. The ANS assay was performed in parallel on liposomes (0.86 mM phosphatidyl choline (PC), 0.24 mM phosphatidyl ethanolamine (PE) and 0.24 mM cardiolipin (CL)) and OMVs, liposomes serving as references of the ANS assay, as described earlier [23]. An increasing concentration in liposomes (1 to 5 µg/µL in Tris-KCl buffer (50 mM Tris-HCl, 200 mM KCl, pH 7.5)) as well as 1 or 2 µL of each OMV solution were incubated with 200 µL of 0.001% ANS (in Tris-KCl buffer) in a black flat-bottom 96-well plate (Corning, Corning, NY, USA). The fluorescence was measured at 512 nm after excitation at 350 nm for 100 ms using a Varioskan plate reader (Thermofisher). Concentration measurements were performed in triplicates, and the mean concentrations in lipids were calculated and used to quantify the OMVs in the experiments involving OMVs. The lipid concentration in REL606 OMVs was found to be 2 mg/mL, while that of miRFP713-OMVs was 0.7 mg/mL.

### 4.5. Quantification of the OMV Contents in Proteins by Bicinchoninic Acid (BCA) Assay

The standard curve of the assay was built using 7 concentrations comprised between 0 and 750 µg/mL of bovine serum albumin (BSA) used as the reference molecule and diluted in phosphate-buffered saline (PBS) solution (0.01 M Na_2_HPO_4_, 0.137 M NaCl, 0.0027 M KCl, 0.0018 M KH_2_PO_4_, pH~7.4). The OMVs’ membrane was dissolved by adding 2 µL of 1.5% Triton- ×100 (Critical Micellar Concentration (CMV): 0.016%) to 150 µL of each OMV solution diluted beforehand in PBS. Then, 0.2 mL of copper sulfate solution (C2284, Sigma) was diluted into 9.8 mL of BCA solution (BCA1-1KT, Sigma). Afterwards, 200 µL of this Cu/BCA mixture was introduced into each well of a transparent flat-bottom 96-well plate (Corning) with 25 µL of each BSA or OMV dilution (both in triplicates), incubated at 37 °C for 30 min, and the absorbance was read at 562 nm using a Varioskan plate reader. The mean concentrations in proteins were calculated and further used in the experiments involving OMVs. The protein concentration of the REL606 OMVs was found to be 2 mg/mL, while that of miRFP713-OMVs was 0.5 mg/mL.

### 4.6. Determination of the OMVs’ Diameter by Dynamic Light Scattering (DLS)

DLS determines the radius of particles within a sample using a coherent monochromatic light source (laser) to light up the sample and measure the angle of the light scattered by the particles. The Stoke–Einstein equation is then used to deduce the diameter of the particles from their diffusion coefficient, knowing the angle of deviation, the viscosity and the temperature of the solution. The DLS of each OMV preparation (10 measurements repeated 3 times) was performed using a Dynapro Nanostar machine (Wyatt technologies, Santa Barbara, CA, USA), which provides the data as a plot of the mean values.

### 4.7. Observation of the OMVs by Transmission Electron Microscopy (TEM) after Negative Stain Drop Technique (DT)

First, 10 µL of OMV solution was added to a glow discharge copper grid coated with a carbon-supporting film (Delta microscopies, Mauressac, France) for 3 min, and the grid was stained for 2 min with 50 µL of 1% phosphotungstic acid (H_3_PW_12_O_40_ in distilled water). The excess of solution was soaked off on a filter paper, and the grid was air-dried. The images were taken under low-dose conditions (<10 e-/Å2) with defocus values between 1.2 and 2.5 μm on a Tecnai 12 LaB6 electron microscope (Stanford, CA, USA) at 120 kV accelerating voltage using a CCD Camera Gatan Orius 1000 (Pleasanton, CA, USA).

### 4.8. Observation of the OMVs’ Proteins Contents

The OMVs’ protein profiles were revealed by sodium dodecyl sulfate–polyacrylamide gel electrophoresis (SDS-PAGE) and Coomassie blue staining using 20 µL volume of each OMV solution solubilized beforehand by the addition of the SDS solution in order to reach a 0.1% final concentration (SDS CMC: 0.15%). After the addition of 5 µL of Laemmli protein sample buffer (containing 350 mM di-thio-threitol and 0.01% Bromophenol blue) into each sample, the mixtures were incubated at 95 °C for 10 min before being loaded into the wells of a 12% SDS-PAGE gel in parallel with a pre-stained standard molecular weight marker (Precision Plus Protein™ All Blue Pre-Stained Protein Standards, Biorad, Hercules, CA, USA). The gel was run for 1 h 20 at 30 mA and 160 V, stained for 30 min in Coomassie blue solution, and then de-stained for 30 min in de-staining solution (10% (*v*:*v*) acetic acid, 70% (*v*:*v*) ethanol, 20% (*v*:*v*) H_2_O) before final incubation in water. A photograph was taken using a Chemidoc imager (Biorad).

### 4.9. Quantification of OmpA in the OMVs after Immunoblotting

For the specific quantification of OmpA, the proteins contained in an SDS-PAGE gel (performed in the same conditions as those described above) were transferred onto a nitrocellulose membrane using transfer buffer (40 mL of 5× Transblot (Biorad), 40 mL of 96% ethanol, 120 mL of H_2_O) in a Trans-Blot Turbo apparatus (Biorad). After the transfer, the membrane was blocked for 30 min at room temperature (RT) in 5% non-fat dry milk dissolved in TBST buffer (9 mM Tris-HCl, 200 mM NaCl, pH 7.5, 0.1% Tween-20); then, it was incubated overnight, at 4 °C, with rabbit polyclonal anti-OmpA immunoglobulins G (IgGs) (Epigentek, Farmingdale, NY, USA) diluted at 1:1000 in 2.5% non-fat milk in TBST under constant agitation. After 2 quick washes and 2 washes of 10 min each in TBST at RT, the membrane was incubated for 3 h at RT with horseradish peroxidase (HRP)-coupled polyclonal anti-rabbit IgGs (Abcam, Cambridge, UK) diluted at 1:10,000 in 2.5% non-fat milk in TBST. After washes with TBST, the signals were developed by chemiluminescence using a 50:50 solution of H_2_O_2_ and luminol and a Chemidoc reader (Biorad). The resulting photographs were processed using the ImageLab 5.1 software.

### 4.10. Mice Experiments

The in vivo experiments were carried out in accordance with the animal experimentation project authorization notified by the French Ministry of Education and Research under APAFiS number #41703-202302232042112 v5. Mandatory authorization of the project was provided by the Comité d’Ethique en Expérimentation Animale CEEA n°12 (Grenoble, France) prior to the carrying out of the experiments. Eight-week-old female BalB/cJRj mice were used. To study the biodistribution of OMVs in mice, experiments were conducted at the optimal imaging platform of the Institute for Advanced Biosciences (IAB) of Grenoble. Before each administration, the mice were anesthetized with 4% isoflurane, and they were maintained under 1.5% isoflurane during the time of fluorescence measurement. Groups of 3 mice are sufficient to ensure that the presence of OMVs in the circulation, and organs can be measured and the kinetics determined. Measurements are not intended to establish statistical or comparative values between groups. For all the in vivo and ex vivo experiments, infrared fluorescence was quantified using a LUMINA III (PerkinElmer) device.

First, we assessed the absence of toxicity of OMVs administered intravenously: 1 mouse was injected with 200 µL of miRFP713-OMVs (equivalent to 140 µg of lipids and 100 µg of proteins) and observed over a 1-day period. Since no suffering was observed, we then went on to measure the fluorescence of the OMVs, when injected intravenously with miRFP713-OMVs (n = 6 mice), by whole-body fluorescence imaging (right and left sides, prone and supine positions) at time 0 h (just before injection), 1 h, 2.5 h, and 5 h p.i. (n = 6 mice, 3 were harvested for isolated direct organ imaging), or 24 h p.i. (n = 3 mice). The 6 mice were injected with 200 µL of miRFP713-OMVs (equivalent to 140 µg of lipids and 100 µg of proteins) + 100 µL of 25 µM BV. Three control mice were mock injected with 200 µL of PBS. The last experiment tracked the fluorescence of OMVs in mice after oral gavage with 200 µL of miRFP713-OMVs (equivalent to 140 µg of lipids and 100 µg of proteins) at Day 0, Day 3 and Day 5, compared to 6 control mice, which did receive an equivalent volume of PBS. For kinetic measurement of fluorescence, 15 µL of blood (into lithium heparin tubes) was collected per mouse from the tail, enabling the sampling of a very low volume, with no consequences for the animals, at 1 min, 5 min, 15 min, 30 min, 1 h, 3 h, 5 h and 24 h p.i. or p.g. of fluorescent OMVs. At the end of the in vivo measurements, the animals were sacrificed by cervical dislocation, their blood was collected, and their organs were isolated for direct ex vivo observation/measurement on the imaging camera LUMINA III (PerkinElmer). Ex vivo fluorescence imaging of miRFP713 was performed on the brain, the heart, the lungs, the liver, the spleen, the pancreas, the kidney, some muscles (quadriceps), some skin from the back, an intestine fragment, the uterus with the ovaries, the adrenal gland, some peritoneal fat after 5 h (n = 3 mice) and 24 h (n = 3 mice) p.i. or after 5 h and 24 h after 1st and 3rd per os administration, respectively, of miRFP713 OMVs. At the end of experiment, the fluorescence was quantified from 10 µL of the plasma obtained after an 8 min centrifugation (7200× *g*) of the blood samples and using a fluorimeter (Varioskan fluorimager, ThermoFisher).

### 4.11. Fluorescence Image Analyses

In vivo fluorescence signals were measured on selected regions of interest (the brain, the heart, the lung, the liver, the spleen, the pancreas, the kidney, the muscle (quadriceps), the skin, an intestine fragment, the uterus with the ovaries, the adrenal gland, some fat). The level of fluorescence background (Bkg) was measured before injection on images of non-injected mice (n = 6 mice/group). In vivo fluorescence signals were expressed as normalized data in reference to signal of t0 values. Ex vivo fluorescence signals were measured on isolated organs and expressed as normalized data in reference to the signal of non-injected mice. The graphs represented the means ± SEM (n = 3). The ex vivo fluorescence background (Bkg) was measured on isolated organs from non-injected mice (n = 3 mice).

### 4.12. Flow Cytometry

Frozen PBMCs were first defrosted in 1× RPMI—20% FBS. After thawing, the cells were counted, pelleted, and dispersed in supplemented 1× RPMI—10% FBS at a concentration of 5 × 10^6^ cells/mL. The cells were incubated with 20 µL of miRFP713-OMVs (equivalent to 10 μg of proteins or 14 μg of lipids) or 20 µL of PBS for 24 h at 37 °C, 5% CO_2_. Labeling with fluorescent antibodies was performed before analysis by flow cytometry using a FACS Canto II. The cells were first gated on a single event (FSC-A/FSC-H and SSC-A/SSC-W), then on live cells (Fixable Viability Stain 510, BD Biosciences, Franklin Lakes, NJ, USA), and finally on morphological parameters (FSC-A/SSC-A). Monocytes and DCs subsets were then identified using antibodies anti-CD11c, -XCR1, -CD40, -MHC II, -CD80 and -SIRPa (BD Biosciences or Miltenyi biotech (Bergisch Gladbach, Germany).

### 4.13. Statistical Analyses

The data were processed using the Excel program. Statistical analyses and figures were performed using Graph Pad prism 9.3.

## 5. Conclusions

As a whole, this study revealed that following gavage, miRFP713-OMVs (1) diffuse rapidly and broadly from the gastro-intestinal tract to the rest of the body and (2) are eliminated from the mouse body through both the intestinal and the urinary tracts; (3) however, they are sufficiently stable in the mouse body and emit enough fluorescence to be efficiently quantified by in vivo imaging of the whole mouse body, ex vivo imaging of isolated mouse organs and fluorometry of blood samples. Flow cytometry analyses confirm that fluorescent OMVs are taken up by monocytes and dendritic cells of myeloid origin in the spleen. This work thus paves the way to further studies that will characterize the cellular interactions and the role of OMVs in their host in healthy or pathological conditions.

## Figures and Tables

**Figure 1 ijms-25-01821-f001:**
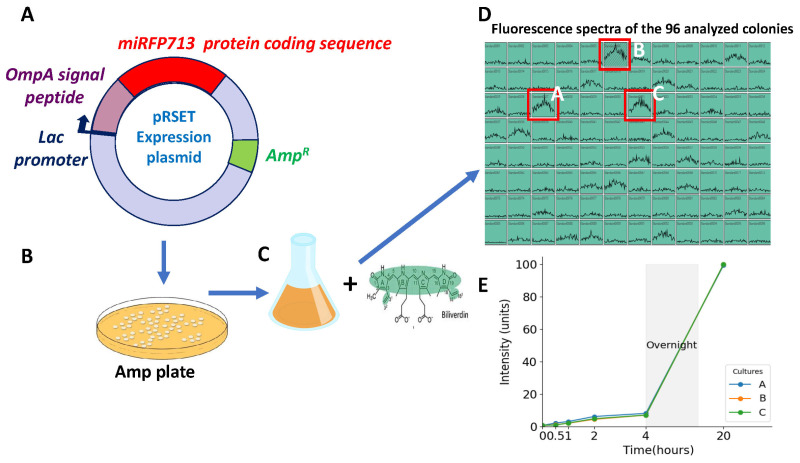
Construction and fluorescence emitted by the REL606 *E. coli* strain expressing miRFP713. (**A**) Schematic representation of the pRSET plasmid constructed to express miRFP713 in C-terminal fusion with the OmpA signal peptide (SP) and under the *lac* promoter. The total fusion OmpA SP-miRFP713 is 1198 nucleotides long. After the selection of ampicillin-resistant colonies on LB-ampicillin agar plates (**B**), 96 colonies were amplified in LB–ampicillin culture medium in the presence of 25 µM biliverdin (**C**). The three cultures with the highest levels of fluorescence (red boxes in **D**) were selected out of the 96 ones using a fluorimeter (Ex. 640 nm, Em. 660 nm) (**D**) and their emitted fluorescence was recorded over 20 h of growth (**E**).

**Figure 2 ijms-25-01821-f002:**
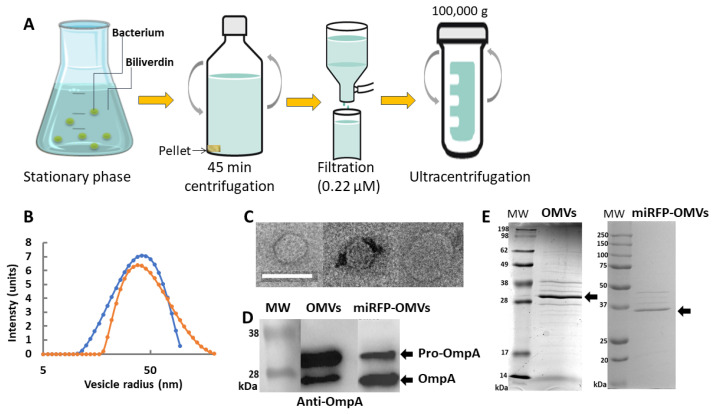
In vitro production and characterization of OMVs secreted from *E. coli* REL606 versus *E. coli* miRFP713-expressing REL606. (**A**) Schematic representation of the various steps of production and purification of fluorescent miRFP713-OMVs versus non-fluorescent OMVs. (**B**) DLS distribution of OMVs (in blue) and miRFP713-OMVs (in orange) radii according to their frequencies. (**C**) Transmission electronic microscopic observations of 3 OMVs. Scale bar: 100 nm. (**D**) Immunoblot detection of the OmpA protein (~27 kDa for the mature protein, ~35 kDa for the pro-protein, black arrow heads) in OMVs (OMVs) versus miRFP713-OMVs (miRFP-OMVs). (**E**) Coomassie blue-stained SDS-PAGE gels of OMVs versus miRFP713-OMVs (miRFP-OMVs) proteins. The OmpA protein is indicated. MW: molecular weight.

**Figure 3 ijms-25-01821-f003:**
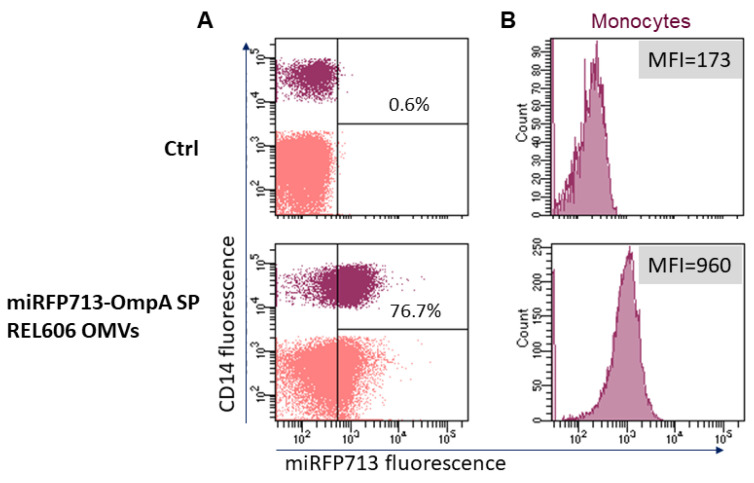
Transfer of miRFP713 fluorescence from miRFP713-OMVs to monocytes using flow cytometry. (**A**) Human Peripheral Blood Mononuclear Cells (in pink) were incubated with miRFP713-OMVs for 24 h. (**B**) Monocytes were identified as CD14-expressing cells (in purple). Monocytes cultured for 24 h without OMVs were used as control (Ctrl). The percentage of miRFP713 positive cells and fluorescent intensities are indicated for monocytes (CD14^+^ cells).

**Figure 4 ijms-25-01821-f004:**
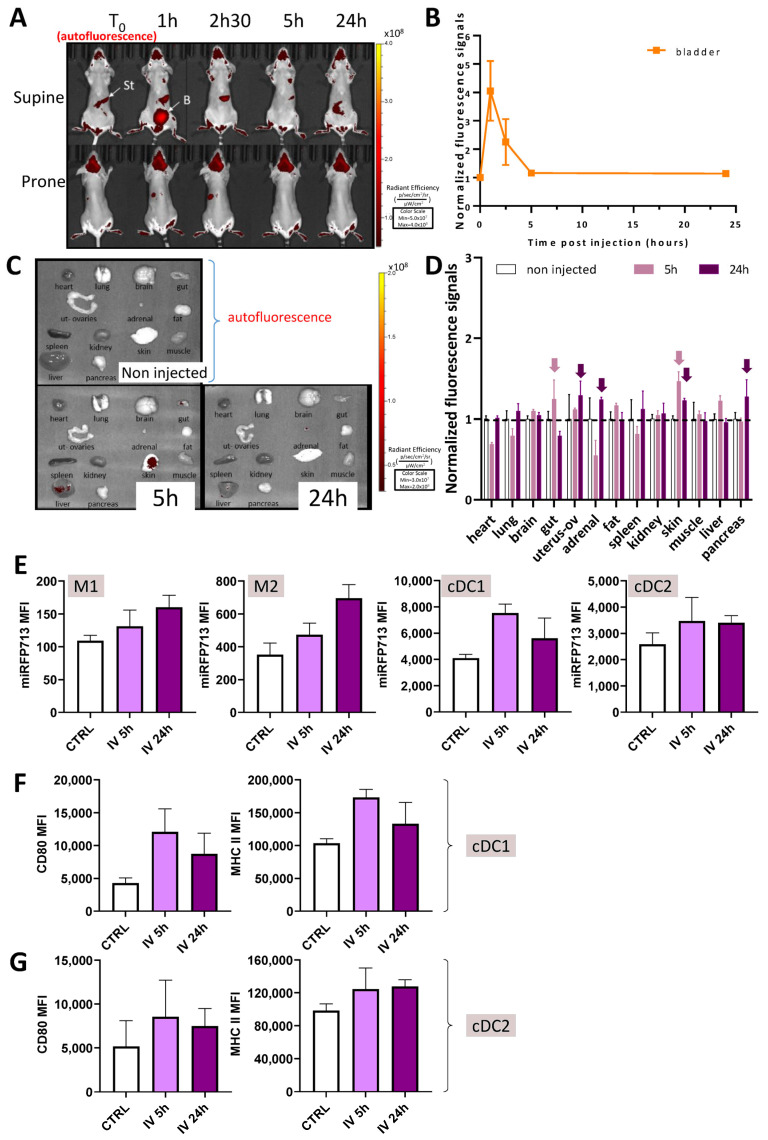
miRFP713-OMVs injected into the tail vein distribute broadly into the whole mouse body where they can be detected in various organs up to 24 h post-injection. (**A**) In vivo whole-body fluorescence imaging of healthy mice over 24 h following intravenous injection of miRFP713-OMVs. The photos are from one mouse representative of the 3 injected ones. In vivo imaging was performed on a LUMINA III (PerkinElmer) device at the time of injection (T = 0), 1 h, 2.5 h, 5 h, and 24 h post-injection. Ex: 620 ± 10 nm; Em: 670 ± 20 nm. St = stomach, B = bladder. (**B**) In vivo levels of fluorescence detected in the bladder of 3 mice at the time of the injection versus 1, 2.5, 5, or 24 h post-injection. (**C**) Ex vivo fluorescence imaging of various organs dissected from mice intravenously injected with miRFP713-OMVs at different times. (**D**) Quantification of ex vivo fluorescence signals after normalization of the signal intensities with those of the non-injected mice. The data represent the mean fluorescence intensity values ± SEM at 5 h (n = 3) versus 24 h (n = 3) post-injection. The dotted horizontal line represents normalization on mouse controls. The arrows indicate results described in the text. (**E**) Flow cytometric analysis of the mean fluorescence intensity (MFI) of miRFP713 spleen monocytes and DCs. Monocytes were defined as viable cells which are CD11b^+^ and CD45^hi^. The subtypes M1 and M2 were then defined by their respective high expression of Ly6c or CD163. Myeloid dendritic cells were defined as live CD11c^+^ MHC class II^hi^ cells; the subtypes cDC1 and cDC2 were then defined by their respective expression of XCR1 or SIRPa (CD172a). (**F**,**G**) The intensity of the activation/maturation markers CD80 as well as the intensity of MHC class II expression is shown on both types of DC. (**E**–**G**) For all cells evaluated by flow cytometry, measurements are shown at T0 (ctrl), 5 h and 24 h after i.v. injection.

**Figure 5 ijms-25-01821-f005:**
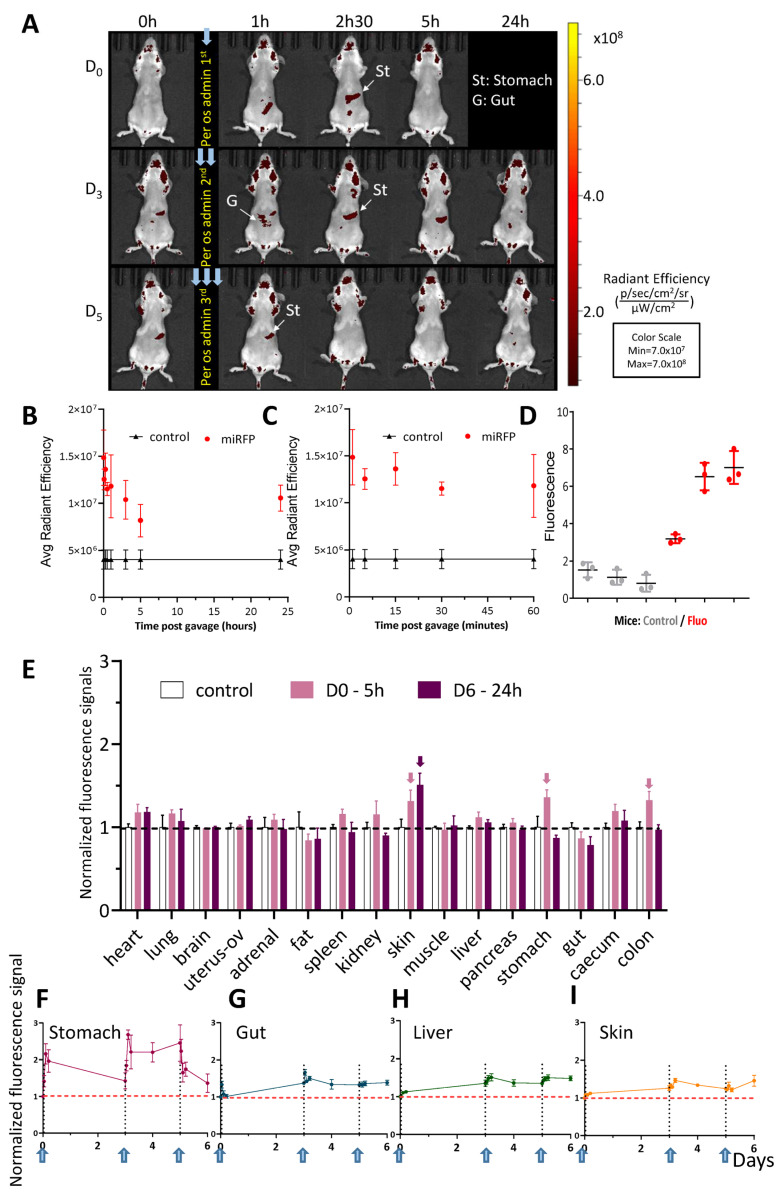
Per os-administered miRFP713-OMVs distribute broadly into the whole mouse body where they can be detected in various organs up to 5 days post-administration. (**A**) In vivo whole-body fluorescence imaging of healthy BalB/C mice, which received 3 per os administrations (Day 0, Day 3, Day 5) of miRFP713-OMVs. The presented data are from one representative mouse out of the 3 BalB/C female mice that were gavaged with miRFP713-OMVs. In vivo imaging was performed on a LUMINA III (PerkinElmer) device at the time of gavage (T = 0), 1 h, 2.5 h, 5 h, and 24 h post-gavage. Ex: 620 ± 10 nm; Em: 670 ± 20 nm. (**B**) Fluorescence signals, as determined using the LUMINA III (PerkinElmer) device, over 24 h in each of 15 µL plasma samples from the 6 healthy mice, which had received per os administration of miRFP713-OMVs or an equivalent volume of PBS. (**C**) Fluorescence signals, as determined using the LUMINA III (PerkinElmer) device, over a period of one hour in each of 15 µL plasma samples from the 6 healthy mice, which had received per os administration of miRFP713-OMVs or an equivalent volume of PBS. (**B**,**C**) The average radial signal (p/s/cm^2^/sr]/[µW/cm^2^) is shown. (**D**) Quantification, using a Varioskan fluorimager (ThermoFisher, Waltham, MA, USA) of the fluorescence levels in the plasma samples from the gavaged mice (red) compared with those detected in the plasma samples from the 3 control-gavaged mice (gray). (**E**) Ex vivo kinetic measurement of the fluorescence signals (mean ± SEM) detected in isolated organs 5 h (n = 3, pink columns) or 24 h (n = 3, purple columns) after per os administration of miRFP713-OMVs. The arrows indicate the results, which are described in the text. The signal intensities (±SEM) were detected kinetically in the organs/tissues: the stomach (**F**), the intestine (referred to “G” (gut) in (**A**) and (**G**)), the liver (**H**), or the skin tissue (**I**) after the first, the second and the third per os administration of miRFP713-OMVs and normalized against the levels of fluorescence detected before administration. Each gavage is symbolized by a vertical dashed line (Days 0, 3, 5), while the horizontal dashed line represents the threshold of fluorescence detection.

## Data Availability

Data is contained within the article and Appendix A.

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
