# Peer review of "Rapid Biodistribution of Fluorescent Outer-Membrane Vesicles from the Intestine to Distant Organs via the Blood in Mice"

_ijms, 2024, doi:10.3390/ijms25031821_

Round 1

Reviewer 1 Report

Comments and Suggestions for Authors

Dear authors,

the work is devoted to the study of the mechanisms of interaction between animals and bacteria - members of the intestinal microbiota. Outer membrane vesicles produced by Gram-negative bacteria and their “fate” in humans and animals certainly deserve the attention of researchers, and the applied method using fluorescence provides a clear picture of the distribution of vesicles after intravenous and oral administration.

However, there are a number of questions about the work.

One of the main questions concerns the introduction and purpose of the research. According to data of PUBMED in 2023 64 reviews devoted to the study of OMVs were published, which formulated the state of current research and issues requiring further study. The list of references does not contain a single reference to works published in 2023. And the introduction does not formulate the specific purpose of the work and does not substantiate the scientific novelty of the research. The fact that OMVs enter the bloodstream from the intestines and circulate through the bloodstream has already been established.

In the chapter materials and methods, paragraph 4.10 raises the most questions.

To facilitate the perception of information, it is necessary to break the paragraph into paragraphs, in each of which clearly indicate how many mice were administered OMVs and how many control mice there were. At what stage was fluorescence measured? Are 3 mice enough to get reliable results?

Blood collection. Was blood taken from each mouse at each point? Did the same apply to control mice? How were the indicators averaged?

Organ retrieval. Was fluorescence measured on whole organs? Immediately after extraction or fixed? How were animals killed?

The list of references must be formatted in accordance with the rules of the journal.

Some abbreviations in the results are not deciphered. Their explanation is given in the materials and methods located at the end of the text, which is not entirely convenient. It is necessary to bring uniformity in the use or non-use of abbreviations. For example, in the text “bileverdin” is written in full or given an abbreviation for no apparent reason.

Figure 1. Was the fluorescence spectrum taken from a bacterial colony or from isolated OMVs? That is, bacteria are also capable of fluorescence? Did you check the glow of the isolated OMVs?

The experiment with bileverdine needs some explanation. Is the increase in fluorescence intensity after 16 hours of incubation due to an increase in the number of bacteria, and not due to the influence of bileverdin? For what purpose was biliverdin administered to mice? Why was it introduced in some experiments and not in others? Have any toxic effects been observed?

Figure 2. If possible, it is necessary to provide a general view of the OMVs and miRFP-OMVs preparations obtained using TEM.

Figure 4. It may be worth dividing the drawing into two parts, since each individual fragment of the drawing is too small and difficult to perceive. An overly long caption also interferes with perception, which may be worth dividing by assigning a separate paragraph to each fragment.

Figure 5. All fragments are of poor quality. The inscriptions are not readable. The fragment F-I signature is incomplete or contains errors.

Author Response

Dear authors,

The work is devoted to the study of the mechanisms of interaction between animals and bacteria - members of the intestinal microbiota. Outer Membrane Vesicles produced by Gram-negative bacteria and their “fate” in humans and animals certainly deserve the attention of researchers, and the applied method using fluorescence provides a clear picture of the distribution of vesicles after intravenous and oral administration.

However, there are a number of questions about the work.

> One of the main questions concerns the introduction and purpose of the research. According to data of PUBMED in 2023 64 reviews devoted to the study of OMVs were published, which formulated the state of current research and issues requiring further study. The list of references does not contain a single reference to works published in 2023. And the introduction does not formulate the specific purpose of the work and does not substantiate the scientific novelty of the research. The fact that OMVs enter the bloodstream from the intestines and circulate through the bloodstream has already been established.

We would like to extend our warmest thanks to the reviewer for his/her constructive criticism and comments on our work.

We have rewritten part of the introduction, with particular emphasis on the limited number of studies in the context of a healthy host in the absence of dysbiosis. We have simplified the characterization of OMVs in the bibliography, lightening the text to focus more on the context of the study and more generally on studies of microbiota and health. Finally, we place greater emphasis on the essential biological role of OMVs in inter-kingdom, inter-species and within holobiont communication.

The changes are annotated in the article's Word document. Finally, we have added recent publications (2023), as requested by the reviewer.

In the chapter materials and methods, paragraph 4.10 raises the most questions.

> To facilitate the perception of information, it is necessary to break the paragraph into paragraphs, in each of which clearly indicate how many mice were administered OMVs and how many control mice there were. At what stage was fluorescence measured? Are 3 mice enough to get reliable results?

Measurement times are specified in the article (please see the revised M&M  and results sections). We have also slightly reworded to clarify.

The lack of relevance of statistical measures to the context of our study is specified in the material and method section.

> Blood collection. Was blood taken from each mouse at each point? Did the same apply to control mice? How were the indicators averaged?

The pharmacokinetic analysis is carried out in 15 µl samples taken from the tail vein of a live animal, on a repeated basis. The aim is to enable measurement without harming the animal or affecting the experiment.

This differs from end-of-experiment measurement in ex vivo analyses, where the blood sample was taken from the retro orbital sinus. The plasma of this blood was used for the measurement using a fluorimeter (not an imager). In this later case, we have used a Varioskan fluorimager (ThermoFisher). We have clarified this point in the document.

All measurements were carried out in the same way on the group of control mice. Of course, measurements on isolated organs were carried out only once, at the end of the 24-hour experiment.

The figures represent average measurements and are described in the article legends.

> Organ retrieval. Was fluorescence measured on whole organs? Immediately after extraction or fixed? How were animals killed?

Yes, fluorescence was measured on complete organs, without preparation, directly ex vivo, without fixation (which would alter measurements). Animals were sacrificed by cervical dislocation. We have clarified this information in the revised draft of the paper.

> The list of references must be formatted in accordance with the rules of the journal.

We have updated the Zotero citation style for International Journal of Molecular Sciences citation in the revised manuscript.

> Some abbreviations in the results are not deciphered. Their explanation is given in the materials and methods located at the end of the text, which is not entirely convenient. It is necessary to bring uniformity in the use or non-use of abbreviations. For example, in the text “bileverdin” is written in full or given an abbreviation for no apparent reason.

We have used the abbreviation BV for biliverdin, except in titles, the abstract and in the legend of figure 1. Abbreviations are introduced when first used in the text of results, if the word is used several times in the manuscript. Conversely, if used only once in the text, we did not use any abbreviation. As an example, even if commonly used by immunologists, we have not used the PAMP abbreviation (for pathogen-associated molecular patterns) because it was mentioned only once, since it was used only once.

> Figure 1. Was the fluorescence spectrum taken from a bacterial colony or from isolated OMVs? That is, bacteria are also capable of fluorescence? Did you check the glow of the isolated OMVs?

Measurement of the fluorescence was performed on the culture wells that comprised the bacterial cells and their medium into which the bacterial cells had released OMVs, .The intrinsic fluorescence of bacterial cells is minimal: most of the detected fluorescence originated from the supernatant containing the OMVs. Supplementary Figure 1 illustrates the fluorescence measurements of excitation and emission spectra of OMVs. Note that during OMV production, the bacterial cells are pelleted to collect only the supernatant containing the OMVs they had produced, followed by a 0.22 µm-filtration step to eliminate any bacterial cell.

> The experiment with bileverdin needs some explanation. Is the increase in fluorescence intensity after 16 hours of incubation due to an increase in the number of bacteria, and not due to the influence of bileverdin? For what purpose was biliverdin administered to mice? Why was it introduced in some experiments and not in others? Have any toxic effects been observed?

As a reminder: the OMVs that we have used are devoid of any bacterial cell (please see above comment): OMV production is completely aseptic. Consequently, mice injected with OMVs are not contaminated with bacterial cells.

Biliverdin is a natural pigment resulting from the degradation of heme. Biliverdin is a natural molecule present in the body of mammals and has been shown to be non-toxic. It plays the role of a cofactor of infrared fluorescent proteins (iRFP) and exacerbates their fluorescence [Bright and stable near-infrared fluorescent protein for in vivo imaging, Nat Biotechnol 2011]. In addition, we verified that biliverdin is not fluorescent. Biliverdin does not stimulate bacterial growth.

We originally planned to add biliverdin only at the time of fluorescence measurements. However, we observed that adding biliverdin during the bacterial culture increased the fluorescence signal. A possible interpretation of this observation is that the cofactor must be in close molecular proximity to the fluorescent protein, as soon as the protein is produced. We did not observe any effect of biliverdin on the number of OMVs produced per bacterial cell.

It is thus very likely that miRFP713 proteins, together with their biliverdin cofactor, are incorporated within OMVs at the time of their production by the bacterial cells.

However, as a security, and to avoid missing any potential interaction between miRFP713 proteins and their biliverdin cofactor we have injected biliverdin together with OMVs into the mice.

As the use of animals is regulated and their number strictly limited to the minimum, we were not able to demonstrate that biliverdin modulates fluorescent measures in vivo. However, as mentioned above, being a natural molecule of mammals, biliverdin is expected to be non-toxic.

> Figure 2. If possible, it is necessary to provide a general view of the OMVs and miRFP-OMVs preparations obtained using TEM.

We are sorry but TEM was not done on the miRFP713-OMVs and we have no general view.

> Figure 4. It may be worth dividing the drawing into two parts, since each individual fragment of the drawing is too small and difficult to perceive. An overly long caption also interferes with perception, which may be worth dividing by assigning a separate paragraph to each fragment.

Figure 4 has been revised, making the axes easier to read. Other modifications are also indicated on the document.

> Figure 5. All fragments are of poor quality. The inscriptions are not readable. The fragment F-I signature is incomplete or contains errors.

Figure 5 has been improved to make the scale readable. In the same way, the legend has also been revised.

Reviewer 2 Report

Comments and Suggestions for Authors

The study investigates the rapid biodistribution of fluorescent outer-membrane vesicles (OMVs) from the intestine to distant organs in mice via the blood. It focuses on Gram-negative bacteria-produced OMVs, previously observed in human blood, potentially facilitating microbiota-host communication. The study optimized OMV signal using miRFP713 protein and biliverdin, and characterized the OMVs' size and composition. In vivo and ex vivo fluorescence imaging tracked OMV distribution post-intravenous injection/oral gavage. I support the publication of this work after minor revision.

1. Section 2.4. From Fig 3A, it seems that some CD14- cells also capture miRFP713 OmpA. The authors should add some discussion about these cell popluations and why they capture miRFP713 OmpA.

2. Section 2.5 The authors should specify what administration method is used for this mouse study.

3. Section 2.5 The author should discuss the different trends between Fig 4E and Fig 4F&G.

4. Section 2.6 The legends and values of Fig. 5 are not readable.

Author Response

The study investigates the rapid biodistribution of fluorescent outer-membrane vesicles (OMVs) from the intestine to distant organs in mice via the blood. It focuses on Gram-negative bacteria-produced OMVs, previously observed in human blood, potentially facilitating microbiota-host communication. The study optimized OMV signal using miRFP713 protein and biliverdin, and characterized the OMVs' size and composition. In vivo and ex vivo fluorescence imaging tracked OMV distribution post-intravenous injection/oral gavage. I support the publication of this work after minor revision.

> Section 2.4. From Fig 3A, it seems that some CD14- cells also capture miRFP713 OmpA. The authors should add some discussion about these cell popluations and why they capture miRFP713 OmpA.

In previous experiments, we labelled OMVs with a lipophilic molecule (DiD, from Molecular Probes, ThermoFischer) and shown that Lymphocytes B, NK cells and few T-cells did incorporate OMVs [Schaack, B.; Hindré, T.; Quansah, N.; Hannani, D.; Mercier, C.; Laurin, D. Microbiota-Derived Extracellular Vesicles Detected in Human Blood from Healthy Donors. Int. J. Mol. Sci. 2022, doi:10.3390/ijms232213787].

Therefore, in the current work, we focused only on the relation between OMVs and monocytes. We have not yet identified the molecules involved in this interaction. There are many potential receptors involved this endocytosis. We have unpublished data (Figure below) showing that other types of blood cells, although in a lesser extent (except for DCs), interacted also with OMVs. These data will be the subject of another publication.

The figure shows the frequency of fluorescent OMV fusion with different types of human blood cells.

> 2. Section 2.5 The authors should specify what administration method is used for this mouse study.

Injection into the tail vein means “intravenous injection”. We have made this clear in the sentence introducing the route of administration.

> 3. Section 2.5 The author should discuss the different trends between Fig 4E and Fig 4F&G.

The reviewer's question shows that our formulation is poorly presented. We have therefore changed Figure 4 to show the interaction between OMVs and cells: M1 or M2 monocytes and cDC1 or cDC2 dendritic cells of myeloid origin in Figure 4E. DC maturation/activation is shown separately in Figures 4F and G. Changes are described in the legend and the text has been modified accordingly.

> 4. Section 2.6 The legends and values of Fig. 5 are not readable.

Figure 5 has been improved to make the scale readable. There was an error in the C part of the figure: the scale was not correct (we now show the accurate details of the first hour's measurements). The ordinate scale in Figures 5 B and C was simplified to ease the reading and the figure legend has been revised accordingly.

We have specified the volume (10 µL) for ex vivo measurement of the OMV fluorescence signal in the plasma.

The in vivo measurement of fluorescence from specific organs is not very accurate. Therefore, animals were killed, their organs isolated, and measurement of their fluorescence was carried out immediately after isolation, without any further organ preparation or fixation. This ex vivo measurement of fluorescence from isolated organs provides more reliable measures of fluorescence linked to the presence of OMVs.

Round 2

Reviewer 1 Report

Comments and Suggestions for Authors

Dear authors, your work was performed at a very high technological level using many modern methods and approaches, which is certainly its advantage, and at the same time makes it quite difficult to comprehend. I am grateful to you for the explanations you provided and the additions made to the text, which allowed me to better understand the tasks you faced and the paths you chose to solve. I would like to note the improved quality of the illustrations and regret the lack of additional photographs obtained using electron microscopy. Control over the purity and quality of extracellular vesicle preparations is absolutely necessary, since impurities that inevitably co-release with vesicles can distort the data obtained, which must be taken into account.

The results presented in the article will certainly be useful to specialists working in this field, and open up opportunities for further study of the interaction of multicellular organisms and intestinal microbiota.

Reviewer 2 Report

Comments and Suggestions for Authors

The authors addressed my comments.